# MAS: Multi-view Ancestral Sampling for 3D motion generation using 2D diffusion

## Abstract

We introduce Multi-view Ancestral Sampling (MAS), a method for generating consistent multi-view 2D samples of a motion sequence, enabling the creation of its corresponding 3D counterpart. While abundant 2D samples are readily available, such as those found in videos, 3D data collection is involved and expensive, often requiring specialized motion-capture systems. MAS leverages diffusion models trained solely on 2D data to produce coherent and realistic 3D motions. This is achieved by simultaneously applying multiple ancestral samplings to denoise multiple 2D sequences representing the same motion from different angles. Our consistency block ensures 3D consistency at each diffusion step by combining the individual generations into a unified 3D sequence, and projecting it back to the original views for the next iteration. We evaluate MAS using 2D pose data from intricate and unique motions, including professional basketball maneuvers, rhythmic gymnastic performances featuring ball apparatus interactions, and horse obstacle course races. In each of these domains, MAS generates diverse, high-quality, and unprecedented 3D sequences that would otherwise require expensive equipment and intensive human labor to obtain. [1]

# 1 Introduction

3D motion generation is an increasingly popular field that has important applications in computer-animated films, video games, virtual reality, and more. One of the main bottlenecks of current approaches is reliance on 3D data, which is typically acquired by actors in motion capture studios or created by professional animation artists. Both forms of data acquisition are not scalable, do not capture in-the-wild behavior, and leave entire motion domains under-explored.

Nevertheless, the ubiquity of video cameras leads to countless high-quality recordings of a wide variety of motions. A possible way to leverage these videos is extracting 3D pose estimations and using them as training data. Yet, the innate ambiguities of monocular 3D pose estimation such as self-occlusions and blurriness of quick motions often lead to infeasible poses and temporal inconsistencies which make the quality of the prediction unsuitable for motion synthesis. Recently, Azadi et al. (2023) and Zhang et al. (2023) incorporated 3D motions estimated from images or videos into motion synthesis applications. The former used them to enrich an existing motion capture dataset and the latter used them as reference motions while learning a physics-based Reinforcement Learning policy. In both cases, the quality issues were bridged using strong priors (either high-quality 3D data or physical simulation), hence remaining limited to the bounds dictated by them.

In this paper, we present Multi-view Ancestral Sampling (MAS), a novel method for utilizing a diffusion model trained on in-the-wild 2D motions to generate 3D motions, including challenging and diverse settings. MAS samples a 3D motion by simultaneously denoising multiple 2D views describing it. At each diffusion denoising step, all views are triangulated into one consistent 3D motion and then projected back to each view. This way we maintain multi-view consistency throughout the denoising process. To further encourage multi-view consistency, we use 3D noise that is projected to each view during sampling.

---

[1] Please watch our supplementary offline web page to see the animated results. Our code, together with the newly extracted datasets, will be made available upon publication.

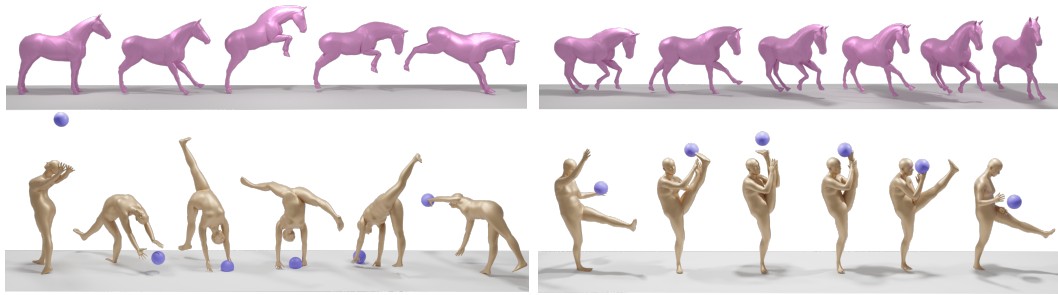

Figure 1: Multi-view Ancestral Sampling (MAS) is using a 2D motion diffusion model to generate novel high-quality 3D motions. This technique enables learning intricate motions from monocular data only.

We show that MAS can sample diverse and high-quality motions, using a 2D diffusion model that was exclusively trained on motions obtained from in-the-wild videos. Furthermore, relying on ancestral sampling allows MAS to generate a 3D motion in a few seconds only, using a single standard GPU. MAS excels in scenarios where acquiring 3D motion capture data is impractical while video footage is abundant (See Figure 1). In such settings, we apply off-the-shelf 2D pose estimators to extract 2D motion from video frames, which are then used to train our diffusion prior. We demonstrate MAS in three domains: (1) professional basketball player motions extracted from common NBA match recordings, (2) horse motions extracted from equestrian contests, and (3) human-ball interactions extracted from rhythmic ball gymnastics performances. Thoese datasets demonstrate motion domains that were previously under-explored due to 3D data scarcity. Our code, together with the newly extracted datasets will be made available upon publication.

## 2 RELATED WORK

**3D Motion Synthesis.** Multiple works explore 3D motion generation using moderate-scale 3D motion datasets such as HumanML3D (Guo et al., 2022), KIT-ML (Plappert et al., 2016) and HumanAct12 (Guo et al., 2020). With this data, synthesis tasks were traditionally learned using Auto-Encoders or VAEs (Kingma & Welling, 2013), (Holden et al., 2016; Ahuja & Morency, 2019; Petrovich et al., 2022; Guo et al., 2022; Tevet et al., 2022). Recently, Denoising Diffusion Models (Sohl-Dickstein et al., 2015; Song & Ermon, 2020) were introduced to this domain by MDM (Tevet et al., 2023), MotionDiffuse (Zhang et al., 2022a), MoFusion (Dabral et al., 2023), and FLAME (Kim et al., 2022). Diffusion models were proven to have a better capacity to model the motion distribution of the data and provided opportunities for new generative tasks. Yet the main limitation of all the above methods is their reliance on high-quality 3D motion capture datasets, which are hard to obtain and limited in domain and scale. In this context, SinMDM (Raab et al., 2023) enabled non-humanoid motion learning from a single animation; PriorMDM (Shafir et al., 2023) and GMD (Karunratanakul et al., 2023) presented fine-tuning and inference time applications for motion tasks with few to none training samples, relying on a pre-trained MDM.

**Monocular Pose Estimation.** Monocular 3D pose estimation is a well-explored field (Kocabas et al., 2020; Shetty et al., 2023; Yu et al., 2023; Shan et al., 2023). Its main challenge is the many ambiguities (e.g. self-occlusions and blurry motion) inherent to the problem. A parallel line of work is pose lifting from 2D to 3D. MotionBERT (Zhu et al., 2023) demonstrates a supervised approach to the task. Some works offer to only use 2D data and learn in an unsupervised manner; Drover et al. (2018) suggest training a 2D discriminator to distinguish between random projections of outputs of a 3D lifting network and the 2D data while optimizing the lifting network to deceive the discriminator; ElePose (Wandt et al., 2021) train a normalizing-flows model on 2D poses and then use it to guide a 3D lifting network to generate 3D poses that upon projection have high probability w.r.t the normalizing-flows model. They add self-consistency and geometric losses and also predict the elevation angle of the 2D pose which is crucial for their success.

**Animal 3D Shape Reconstruction.** The recent MagicPony (Wu et al., 2023) estimates the pose of an animal given a single image by learning a per-category 3D shape template and per-instance

skeleton articulations, trained to reconstruct a set of 2D images upon rendering. Yao et al. (2023) suggest a method for improving the input images with occlusions/truncation via 2D diffusion. Then, they use a text-to-image diffusion model to guide 3D optimization process to obtain shapes and textures that are faithful to the input images.

**Text to 3D Scene Generation.** DreamFusion (Poole et al., 2022) and SJC (Wang et al., 2022), introduced guidance of 3D content creation using diffusion models trained on 2D data. Poole et al. (2022) suggest SDS, a method for sampling from the diffusion model by minimizing a loss term that represents the distance between the model's distribution and the noised sample distribution. They suggest to harness SDS for 3D generation by repeatedly rendering a 3D representation (mostly NeRF (Mildenhall et al., 2020) based) through a differentiable renderer, noising the resulting images using the forward diffusion, get a correction direction using the diffusion model, and then back-propagate gradients to update the 3D representation according to the predicted corrections. Although promising, their results are of relatively low quality and diversity and suffer from slow inference speed, overly saturated colors, lack of 3D consistency, and heavy reliance on text conditioning. Follow-up works elaborate on the concept and suggest methods for improvement. Magic3D (Lin et al., 2023) adopt coarse-to-fine optimization strategy and improve design choices; Fantasia3D (Chen et al., 2023) suggest starting by optimizing a geometric representation using normal maps rendered from it and fed into a text-to-image diffusion model, and then optimize the surface material; ProlificDreamer (Wang et al., 2023c) suggest modeling the 3D scene as a random variable, and using a particle-based variational inference approach, which enables the generation of diverse scenes; HIFA (Zhu & Zhuang, 2023) apply a DDIM (Song et al., 2022) sampling loop inside each optimization iteration, add depth and density priors and improve design choices such as timestep scheduling; DreamTime (Huang et al., 2023) thoroughly explores timestep scheduling and weighting and suggests a monotone timestep schedule and a weight function that is divided into 3 sections - coarse, content, and detailed; Hertz et al. (2023) suggested the Delta Denoising Score to avoid mode-collapse in image editing applications. In a similar context, Instruct-NeRF2NeRF (Haque et al., 2023) edit a NeRF by gradually editing its source multi-view image dataset during training, using an image diffusion model.

## 3 PRELIMINARY

**Diffusion Models and Ancestral Sampling.** Diffusion models are generative models that learn to gradually transform a predefined noise distribution into the data distribution. For the sake of simplicity, we consider the source distribution to be Gaussian. The forward diffusion process is defined by taking a data sample and gradually adding noise to it until we get a Gaussian distribution. The diffusion denoising model is then parameterized according to the reverse of this process, i.e. the model will sample a random Gaussian sample and gradually denoise it until getting a valid sample.

Formally, the forward process is defined by sampling a data sample $x_0 \sim q(x_0)$ and for $t$ in $1, ..., T$, sampling $x_t \sim q(x_t|x_{t-1}) = \mathcal{N}(x_t; \sqrt{1-\beta_t}x_{t-1}, \beta_t I)$, until getting to $x_T$, which has a gaussian distribution $x_T \sim q(x_T) = \mathcal{N}(x_T; 0, I)$.

The reverse process, also called **ancestral sampling**, is defined by sampling a random gaussian noise $x_T \sim p_\phi(x_T) = \mathcal{N}(x_T; 0, I)$ and then for $t$ in $T, t-1, ..., 1$, sampling $\hat{x}_{t-1} \sim p_\phi(\hat{x}_{t-1}|x_t)$, until getting to $\hat{x}_0$, which should ideally approximate the data distribution. The model posterior $p_\phi(x_{t-1}|x_t)$ is parameterized by a network $\mu_\phi(x_t, t)$: $p_\phi(x_{t-1}|x_t) = q(x_{t-1}|x_t, x_0 = \mu_\phi(x_t; t)) = \mathcal{N}(x_{t-1}; \mu_\phi(x_t, t), \sigma_t^2 I)$ i.e. the new network predicts a mean denoising direction from $x_t$ which is then used for sampling $x_{t-1}$ from the posterior distribution derived from the forward process. $\mu_\phi$ is further parameterized by a network $\epsilon_\phi$ that aims to predict the noise embedded in $x_t$:

$$\mu_\phi(x_t, t) = \frac{1}{\sqrt{\alpha_t}}\left(x_t - \frac{\beta_t}{\sqrt{1-\bar{\alpha}_t}}\epsilon_\phi(x_t, t)\right) \qquad (1)$$

Now, when optimizing the usual variational bound on negative log-likelihood, it simplifies to,

$$\mathcal{L}(\phi) = \mathbb{E}_{t \sim \mathcal{U}(0,1), \epsilon \sim \mathcal{N}(\mathbf{0}, \mathbf{I}), x_0 \sim q(x_0)}\left[w(t)\|\epsilon_\phi(\alpha_t x_0 + \sigma_t \epsilon; t) - \epsilon\|_2^2\right] \qquad (2)$$

which is used as the training loss. We approximate this loss by sampling $t, \epsilon, x_0$ from their corresponding distributions and calculating the loss term.

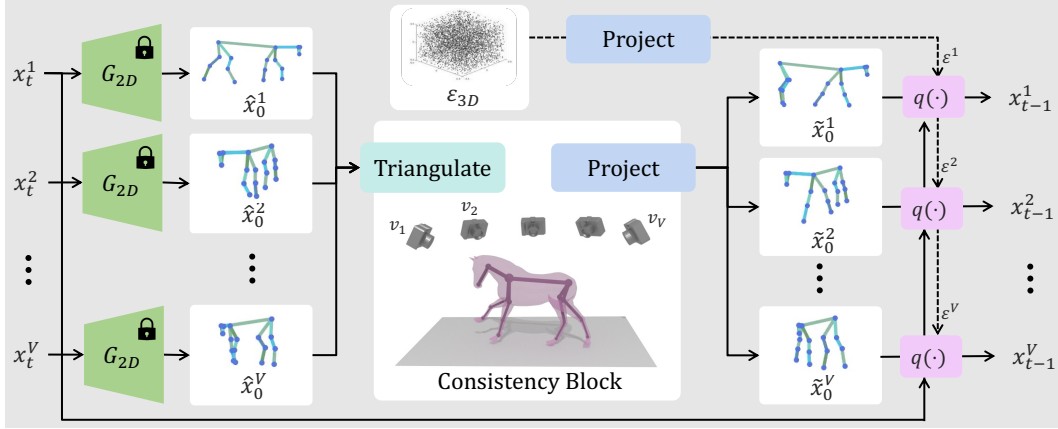

Figure 2: The figure illustrates an overview of MAS, showing a multi-view denoising step from the 2D sample collection $x_t^{1:V}$ to $x_{t-1}^{1:V}$, corresponding to camera views $v_{1:V}$. Denoising is performed by a pre-trained 2D motion diffusion model $G_{2D}$. At each such iteration, our *Consistency Block* triangulates the motion predictions $\hat{x}_0^{1:V}$ into a single 3D sequence and projects it back onto each view ($\tilde{x}_0^{1:V}$). To encourage consistency in the model's predictions, we sample 3D noise, $\epsilon_{3D}$ and project it to the 2D noise set $\epsilon^{1:V}$ for each view. Finally, we sample $x_{t-1}^{1:V}$ from $q\left(x_{t-1}^{1:V} | x_t^{1:V}, \tilde{x}_0^{1:V}\right)$.

**Data Representation.** A motion sequence is defined on top of a character skeleton with $J$ joints. A single character pose is achieved by placing each joint in space. Varying the character pose over time constructs a motion sequence. Hence, we denote a 3D motion sequence, $X \in \mathbb{R}^{L \times J \times 3}$, with $L$ frames by the $xyz$ location of each joint at each frame. Note that this representation is not explicitly force fixed bone length. Instead, our algorithm will do so implicitly. Additionally, This formulation allows us to model additional moving objects in the scene (e.g. a ball or a box) using auxiliary joints to describe their location.

Considering the pinhole camera model [2], we define a camera-view $v = (R_v, \tau_v, f_v)$ by its rotation matrix $R_v \in \mathbb{R}^{3 \times 3}$, translation vector $\tau_v \in \mathbb{R}^3$ and the focal length $f_v$ given in meters. Then, a 2D motion, $x^v = p(X, v) \in \mathbb{R}^{L \times J \times 2}$, from camera-view $v$, is defined as the perspective projection $p$ of $X$ to $v$ such that each joint at each frame is represented with its $uv$ coordinates of the camera space.

In order to drive 3D rigged characters (as presented in the figures of this paper) we retrieve 3D joint angles from the predicted 3D joint positions of $X$ using SMPLify (Bogo et al., 2016) optimization for human characters, and Inverse-Kinematics optimization for the non-humanoid characters (i.e. horses).

## 4 METHOD

Our goal is to generate 3D motion sequences using a diffusion model trained on monocular 2D motions. This would enable 3D motion generation in the absence of high-quality 3D data, by leveraging the ubiquity of monocular videos describing those scenes. To this end, we introduce Multi-view Ancestral Sampling (MAS), a method that simultaneously generates multiple views of a 3D motion via ancestral sampling. MAS maintains consistency between the 2D motions in all views to construct a coherent 3D motion at each denoising step. A single MAS step is illustrated in Figure 2.

First, we extract 2D pose estimations from in-the-wild videos and use them to train a 2D diffusion model $\hat{x}_0 = G_{2D}(x_t)$, based on the MDM (Tevet et al., 2023) architecture that predicts the clean 2D motion, $\hat{x}_0$ at each denoising step (See Figure 3).

MAS is then able to sample 3D motions from $G_{2D}$ as follows. MAS simultaneously applies a DDPM ancestral sampling loop on multiple 2D motions, which represent views of the same 3D motion from $V$ different camera angles. At each denoising step $t$, we get a set of noisy views $x_t^{1:V}$

---
[2]https://en.wikipedia.org/wiki/3D_projection#Perspective_projection

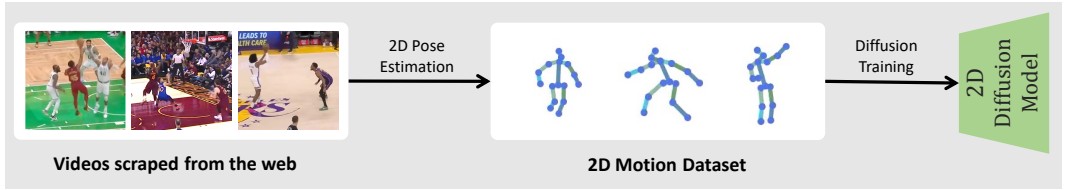

Figure 3: **Preparations.** The motion diffusion model used for MAS is trained on 2D motion estimations of videos scraped from the web.

as input and predict clean samples $\hat{x}_0^{1:V} = G_{2D}(x_t^{1:V})$. Then, the *Consistency Block* is applied in two steps: (1) Triangulation: find a 3D motion $X$ that follows all views as closely as possible. (2) Reprojection: project the resulting 3D motion to each view, getting $\tilde{x}_0^{1:V}$, which we can think of as a multiview-consistent version of the predicted denoising direction. Finally, we can sample the next step $x_{t-1}^{1:V}$ from the backward posterior $x_{t-1}^{1:V} \sim q\left(x_{t-1}|x_t, \tilde{x}_0^{1:V}\right)$. Repeating this denoising process up to $t = 0$ yields the same motion sequence generated from $V$ views. Those sequences are triangulated to construct a 3D motion which is the output of MAS. This sampling process is detailed in Algorithm 1 (Appendix). The remainder of this section describes the monocular data collection and diffusion pre-training (4.1), followed by a full description of MAS building blocks (4.2).

## 4.1 PREPARATIONS

**Data Collection.** We collect videos from various sources — NBA videos, horse jumping contests, and rhythmic gymnastics contests. We then apply multi-person and object tracking using off-the-shelf models to extract bounding boxes. Subsequently, we use other off-the-shelf models for 2D pose estimation to get 2D motions. Implementation details are in Section 6. We build on the fact that 2D pose estimation is a well-explored topic, with large-scale datasets that can be easily scaled as manual annotations are much easier to obtain compared to 3D annotation which usually requires a motion capture studio.

**2D Diffusion Model Training.** We follow Tevet et al. (2023) and train the unconditioned version of the Motion Diffusion Model (MDM) with a transformer encoder backbone for each of the three datasets separately. We boost the sampling of MDM by a factor of 10 by learning 100 diffusion steps instead of the original 1000.

## 4.2 MULTI-VIEW ANCESTRAL SAMPLING

We would like to construct a way to sample a 3D motion using a model that generates 2D samples. First, we observe that a 3D motion is uniquely defined by 2D views of it from multiple angles. Second, we assume that our collected dataset includes a variety of motions, from multiple view-points, and deduce that our 2D diffusion model can generalize for generating multiple views of the same 3D motion, for a wide variety of 3D motions. Thus, we aim to generate multiple 2D motions that represent multiple views of the same 3D motion, from a set of predefined view-points.

**Ancestral Sampling for 3D generation.** As described in Section 3, diffusion models are designed to be sampled using gradual denoising, following the ancestral sampling scheme. Hence, we design MAS to generate multiple 2D motions via ancestral sampling, while guiding all views to be multiview-consistent. Formally, we take a set of $V$ views, distributed evenly around the motion, with elevation angle distribution heuristically picked for each dataset. Then, for a each view $v$ we initialize $x_T^v$, and for $t = T, ..., 1$ transform $x_t^v$ to $x_{t-1}^v$ until getting a valid 2D motion $x_0^v$ for each view. We choose to generate all views concurrently, keeping all views in the same diffusion timestep throughout the process.

In every denoising step we receive $x_t^{1:V} = \left(x_t^1, ..., x_t^V\right)$. We derive the clean motion predictions by applying the diffusion model in each view $\hat{x}_0^v := \frac{x_t^v - \sqrt{1-\bar{\alpha}_t}\epsilon_\phi(x_t^v)}{\sqrt{\bar{\alpha}_t}}$, getting $\hat{x}_0^{1:V} = \left(\hat{x}_0^1, ..., \hat{x}_0^V\right)$. We apply our multi-view Consistency Block to find multi-view consistent denoising direction $\tilde{x}_0^{1:V}$ that approximates the predicted motions $\hat{x}_0^{1:V}$. We then use the resulting motions $\tilde{x}_0^{1:V}$ as the denoising direction by sampling $x_{t-1}^v$ from $q\left(x_{t-1}^v|x_t^v, x_0 = \tilde{x}_0^v\right)$, and outputting $x_{t-1}^{1:V} = \left(x_{t-1}^1, ..., x_{t-1}^V\right)$.

MAS can be extended to support dynamic camera-view along sampling instead of fixed ones as detailed in Appendix D. Since this is not empirically helpful for our application, we leave it out of our scope.

**Multi-view Consistency Block** As mentioned, the purpose of this block is to transform multiview motions $\hat{x}_0^{1:V}$ into multiview-consistent motions $\tilde{x}_0^{1:V}$ that are as similar as possible. We achieve this by finding a 3D motion $X$ that when projected to all views, it resembles the multiview motions $\hat{x}_0^{1:V}$ via *Triangulation*. We then return projections of $X$ to each view $\tilde{x}_0^{1:V} = (p(X, 1), ..., p(X, V))$, as the multiview-consistent motions.

**Triangulation.** We calculate $X$ via optimization to minimize the difference between projections of $X$ to all views and the multiview motion predictions $\hat{x}_0^{1:V}$:

$$X = \arg\min_{X'} \|p(X', 1{:}V) - \hat{x}_0^{1:V}\|_2^2 = \arg\min_{X'} \sum_{v=1}^{V} \|p(X', v) - \hat{x}_0^v\|_2^2$$

For faster convergence, we initialize $X$ with the optimized results from the previous sampling step. This way the process can also be thought of progressively refining $X$ but we wish to emphasize that the focus remains the ancestral sampling in the 2D views.

**3D Noise.** In addition to the consistency block, that enforces 3D consistency on the multi-view predictions $\tilde{x}_0^{1:V}$, we would like the noise to be consistent between views as well. To this end, we design a new noise sampling mechanism that will (1) keep Gaussian distribution for each view, and (2) maintain 3D consistency between the views.

We start by sampling 3D noise $\varepsilon_{3d} \sim \mathcal{N}(0, I)$ $(\varepsilon_{3d} \in \mathbb{R}^{L \times J \times 3})$. Projecting this noise to each view using the perspective projection $p$ will break the Gaussian assumption, hence, we use orthographic projection instead, which preserves Gaussian distribution for each view (see Appendix C,1), and observes $O(1/d)$ error compared to the perspective projection, where $d$ is the distance between the camera and the object (see Appendix C,2). We then use the resulting distribution for sampling the initial noise $x_T$ and when sampling $x_{t-1} \sim q(x_{t-1}|x_t, x_0 = p(X))$.

## 5 METHOD DISCUSSION

In this section, we discuss the properties of MAS, contextualizing it within the landscape of recent advancements in the text-to-3D domain.

**Ancestral sampling.** MAS is built upon the ancestral sampling process. This means that the model is used in its intended way over in-domain samples. This is in contrast to SDS-based methods (Poole et al., 2022) which employ a sampling scheme that uses the forward diffusion to noise images rendered from a 3D representation that is only partially optimized. This can lead to out-of-distribution samples, particularly ones in the early timesteps, where the noise is rather large. This phenomenon is also addressed by Wang et al. (2022) and Huang et al. (2023), who suggest heuristics to alleviate the out-of-distribution problem but do not fundamentally solve it. Furthermore, most SDS-based methods sample $x_t$ independently in each iteration, which may lead to a high variance in the correction signal. Contrarily, using ancestral sampling has, by definition, a large correlation between $x_t$ and $x_{t-1}$, which leads to a more stable process and expressive results. Since MAS is sampling-based, it naturally models the diversity of the distribution, while optimization-based methods often experience mode-collapse or divergence, as addressed by Poole et al. (2022). It is worth noting that SDS is a clever design for cases where ancestral sampling cannot be used.

**Multi-view stability.** MAS simultaneously samples multiple views that share the same timestep at each denoising step. SDS-based methods typically use a single view in each optimization step, forcing them to make concessions such as small and partial corrections to prevent ruining the 3D object from other views. This also leads to a state where it is unknown which timestep to choose, since only partial denoising steps were applied (also shown by Huang et al. (2023)). MAS avoids such problems since the multiple view denoising steps are applied simultaneously. It allows us to apply full optimization during the triangulation process. Hence, at the end of the $i$'th iteration, the motion follows the model's distribution at timestep $T - i$. This alleviates the need for timestep scheduling while dramatically decreasing inference time.

**3D noise consistency.** MAS usage of a noise distribution projected from 3D noise onto each view, boosts multiview-consistency in the model's predictions and greatly benefits the quality and diversity

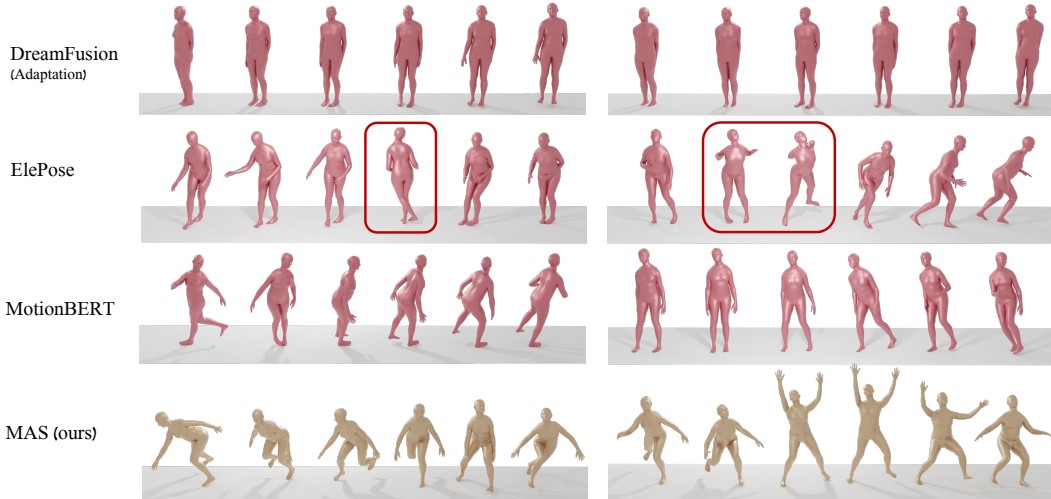

Figure 4: Generated motions by MAS compared to ElePose (2021), MotionBert (2023), and the motion adaptation of DreamFusion (2022). While MotionBert and DreamFusion generated dull motions with limited movement, ElePose generations are jittery and often include invalid poses (Red rectangles).

of the generated motions. SDS-based methods sample uncorrelated noise in the different views, and this inconsistency among the views leads to slower convergence or even divergence.

## 6 EXPERIMENTS

In order to demonstrate the merits of our method, we apply MAS on three different 2D motion datasets. Each dataset addresses a different motion aspect that is under-represented in existing 3D motion datasets (See table 1). (1) The NBA players' performance dataset demonstrates motion generation in domains of human motions that are poorly covered by existing datasets. (2) The horse show-jumping contests dataset shows generation in a domain that has almost no 3D data at all and has a completely different topology. Finally, (3) the rhythmic-ball gymnastics dataset shows that our method opens the possibility to model interactions with dynamic objects. All the datasets, along with our code, will be made available upon publication.

### 6.1 DATA COLLECTION

**NBA videos.** We collected about $10K$ videos from the NBA online API[3]. We then applied multi-person tracking using ByteTrack (Zhang et al., 2022b), and AlphaPose (Fang et al., 2022) for 2D human pose estimation (based on the tracking results). We finally processed and filtered the data by centering the people, filtering short motions, crowd motions, and motions of low quality, splitting discontinuous motions (caused typically by tracking errors), mirroring, and applying smoothing interpolations.

**Horse jumping contests.** We collected 3 horse jumping contest videos (around 2-3 hours each) from YouTube.com. We then apply YoloV7 (Wang et al., 2023a) for horse detection and tracking and VitPose (Xu et al., 2022) trained on APT-36K (Yang et al., 2022) for horse pose estimation. The post-processing pipeline was similar to the one described above.

**Rhytmic-ball gymnastics**. We used the Rhythmic Gymnastics Dataset (Zeng et al., 2020) to get 250 videos, about 1.5 minutes long each, of high-standard international competitions of rhythmic gymnastics performance with a ball. We followed the pipeline described for NBA videos to obtain athletes' motions and also use YoloV7 for detecting bounding boxes of sports balls. We take the

---

[3]https://github.com/swar/nba_api

Table 1: **2D Datasets.** Details of the three in-the-wild datasets, collected to demonstrate MAS generation capabilities in under-explored motion domains.

| Dataset Name | Subject | #Samples | Length Range | Average Length | FPS |
|---|---|---|---|---|---|
| NBA videos | Humans | $\sim 60K$ | 4s-16s | $\sim 6s$ | 30 |
| Horse jumping contests | Horses | $\sim 2K$ | 3s-40s | $\sim 7s$ | 20 |
| Rhythmic ball gymnastics | Humans + Ball | $\sim 500$ | 10s-120s | $\sim 81s$ | 20 |

closest ball to the athlete at each frame and add the center of the bounding box as an additional "joint" in the motion representation.

All motions are presented as $x \in \mathbb{R}^{L \times J \times 2}$ as was detailed in Section 3, where NBA is using the AlphaPose body model with 16 joint, horses represented according to APT-36K with 17 joints and the gymnastics dataset is represented with the COCO body model (Lin et al., 2015) with 17 joints plus additional joint for the ball. All 2D pose predictions are accompanied by confidence predictions per joint per frame which are used in the diffusion training process.

## 6.2 IMPLEMENTATION DETAILS

Our 2D diffusion model is based on MDM (Tevet et al., 2023), composed of a transformer encoder with 6 attention layers of 4 heads and a latent dimension of 512. This backbone supports motions with variable length in both training and sampling, which makes MAS support them as well. To mitigate some of the pose prediction errors, we mask low-confidence joint predictions from the diffusion loss during training. We used an ADAM optimizer with 0.1 lr for training and cosine noise scheduling. We learn 100 diffusion steps instead of 1000 which accelerate MAS 10-fold without compromising the quality of the results. We observe that MAS performs similarly for any $V \geq 3$ and report 5 camera views along all our experiments. The camera views $v_{1:V}$ are fixed through sampling, surrounding the character and sharing the same elevation angle. The azimuth angles evenly spread around $[0, 2\pi]$. Generating a sample with MAS takes less than 10 seconds on a single NVIDIA GeForce RTX 2080 Ti GPU.

## 6.3 EVALUATION

Here we explore the quality of the 3D motions generated by our method. Usually, we would compare the generated motions to motions sampled from the dataset. In our case, we do not have 3D data so we must introduce a new way to evaluate the 3D generated motions. For that sake, we rely on the assumption that a 3D motion is of high-quality if and only if all 2D views of it are of high-quality. Consequently, we suggest taking random projections of the 3D motions and comparing them with our 3D data. More specifically, we generate a set of 3D motions, with lengths sampled from the data distribution, then sample a single angle for every motion with yaw drawn from $\mathcal{U}[0, 2\pi]$ and a constant pitch angle fitted for each dataset. We project the 3D motion to the sampled angle using perspective projection, from a constant distance (also fitted for each dataset) and get a set of 2D motions.

Finally, we follow common evaluation metrics (Raab et al., 2022; Tevet et al., 2023) used for assessing unconditional generative models: *FID* measures Fréchet inception distance between the tested distribution and the test data distribution; *Diversity* measures the variance of generated motion in latent space; *Precision* measures the portion of the generated data that is covered by the test data; *Recall* measures the portion of the test data distribution that is covered by the measured distribution. Those metrics are calculated in latent space. Hence, we follow the setting suggested by (Guo et al., 2020) and train a VAE-based evaluator for each dataset. This setting become the de-facto standard for motion evaluation (Petrovich et al., 2022; Tevet et al., 2023; Guo et al., 2022; Jiang et al., 2023). We evaluate over $1K$ random samples and repeat the process 10 times to calculate the standard deviation.

Table 2 shows that MAS results are comparable to the diffusion model in use, which marks a performance upper bound in 2D. In addition, MAS suffers from a mode-collapse without the 3D noise feature. Lastly, we evaluate a motion adaptation of DreamFusion (Poole et al., 2022) and show that

Table 2: We compare MAS to a motion DreamFusion (Poole et al., 2022) adaptation according to the quality of 2D projections of the generated motions. The performance of the pre-trained 2D diffusion model used by both MAS and DreamFusion serves as an upper bound, yet MAS achieves comparable results. Our ablations show that MAS performs best with as few as 5 views (ours), and 3D noise is crucial for convergence. gray marks mode-collapse (Recall$<10\%$), **bold** marks best results otherwise. '$\rightarrow$' means results are better when the value is closer to the real distribution.

| | FID$\downarrow$ | Diversity$\rightarrow$ | Precision$\uparrow$ | Recall$\uparrow$ |
|---|---|---|---|---|
| Ground-Truth | $1.05^{\pm.02}$ | $8.97^{\pm.05}$ | $0.73^{\pm.01}$ | $0.73^{\pm.01}$ |
| 2D Diffusion Model | $5.23^{\pm.13}$ | $9.70^{\pm.08}$ | $0.44^{\pm.02}$ | $0.78^{\pm.01}$ |
| MAS (Ours) | $\mathbf{5.38^{\pm.06}}$ | $\mathbf{9.47^{\pm.06}}$ | $\mathbf{0.50^{\pm.01}}$ | $0.60^{\pm.01}$ |
| with 2 views (120°) | $6.87^{\pm.14}$ | $9.99^{\pm.06}$ | $0.35^{\pm.01}$ | $\mathbf{0.80^{\pm.01}}$ |
| - 3d noise | $17.40^{\pm.12}$ | $6.67^{\pm.07}$ | $0.93^{\pm.01}$ | $0.01^{\pm 2.6e-03}$ |
| DreamFusion (2022) | $66.38^{\pm1.24}$ | $8.25^{\pm.16}$ | $0.33^{\pm.08}$ | $0.17^{\pm.13}$ |

Table 3: **Comparison with pose lifting.** MAS outperforms state-of-the-art unsupervised lifting methods. Furthermore, lifting methods are failing short when evaluated from the side view $(\mathcal{U}\left(\frac{\pi}{4}, \frac{3\pi}{4}\right))$, while the performance of MAS remains stable '$\rightarrow$' means results are better when the value is closer to the real distribution (8.97 for Diversity); **bold** marks best results.

| | FID$\downarrow$ | | Diversity$\rightarrow$ | | Precision$\uparrow$ | | Recall$\uparrow$ | |
|---|---|---|---|---|---|---|---|---|
| View Angles | All | Side | All | Side | All | Side | All | Side |
| ElePose (2021) | $10.76^{\pm.45}$ | $18.28^{\pm.33}$ | $9.72^{\pm.05}$ | $\mathbf{8.98^{\pm.06}}$ | $0.28^{\pm.02}$ | $0.26^{\pm.02}$ | $0.58^{\pm.03}$ | $0.17^{\pm.01}$ |
| MotionBert (2023) | $30.22^{\pm.26}$ | $36.89^{\pm.40}$ | $9.57^{\pm.09}$ | $8.67^{\pm.08}$ | $0.04^{\pm 4e-03}$ | $0.03^{\pm.01}$ | $0.34^{\pm.04}$ | $0.15^{\pm.04}$ |
| MAS (Ours) | $\mathbf{5.38^{\pm.06}}$ | $\mathbf{5.43^{\pm.11}}$ | $\mathbf{9.47^{\pm.06}}$ | $9.49^{\pm.04}$ | $\mathbf{0.50^{\pm.01}}$ | $\mathbf{0.51^{\pm.02}}$ | $\mathbf{0.60^{\pm.01}}$ | $\mathbf{0.61^{\pm.01}}$ |

it performs poorly. This adaptation optimizes the 3D motion $X$ directly through 10K iterations for each sample, each updates $X$ using SDS gradients of the same diffusion model used for MAS.

Table 3 shows that the generated motions by MAS outperform state-of-the-art pose lifting methods, both the supervised MotionBERT (Zhu et al., 2023) and the unsupervised ElePose (Wandt et al., 2021). Although these methods are not generative per se, we consider lifted motions from 2D motions sampled from the training data as generated samples. Since we sample a uniform angle around the generated motions, we often sample angles that are similar to the lifted angle, which was given as an input to the lifters but not to MAS. When evaluating from the side view, (angle $\sim \mathcal{U}\left(\frac{\pi}{4}, \frac{3\pi}{4}\right)$) we see the clear degradation of the lifting methods while MAS keep the same performance. Figure 6 demonstrates the quality of MAS compared to DreamFusion, MotionBERT, and ElePose.

# 7 CONCLUSIONS, LIMITATIONS AND FUTURE WORK

In this paper, we introduced MAS, a generative method designed for 3D motion synthesis. We showed that high-quality 3D motions can be sampled from a diffusion model trained on 2D data only. The essence of our method lies in its utilization of a multiview diffusion ancestral sampling process, where each denoising step contributes to forging a coherent 3D motion sequence. Remarkably, our experiments show that MAS excels with in-the-wild videos, enabling it to produce motions that are otherwise exceedingly challenging to obtain through conventional means.

Our method could also be employed in additional domains such as multiperson interactions, other animal motions and with recent developments in tracking of "any" object (Wang et al., 2023b), we wish to push the boundaries of data even further.

Our method inherits the limitations of the 2D data it is using and thus cannot naively predict global position, or apply textual control. We leave extending the data acquisition pipeline to support such features to future work. Finally, we hope the insights introduced in this paper can also be utilized in the text-to-3D field and other applications.

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

## APPENDIX

## A THE MAS ALGORITHM

Algorithm 1 describes the MAS sampling process.

---

**Algorithm 1** Multi-view Ancestral Sampling (MAS)

---
1: **Sample camera views:** $v_{1:V} \sim \mathcal{V}$
2: **Initialize and noise project:** $x_T^{1:V} \xleftarrow{\text{project}} \varepsilon_{3D} \sim \mathcal{N}(0, I)$
3: **for** $t = T, T-1, ..., 0$ **do**
4:      $\hat{x}_0^{1:V} = G_{2D}\left(x_t^{1:V}\right)$
5:      **Triangulate:** $X = \underset{X'}{\arg\min} \, ||p(X', v_{1:V}) - \hat{x}_0^{1:V}||_2^2$          $\triangleright X, \varepsilon_{3D} \in \mathbb{R}^{L \times J \times 3}$
6:      **Back-project:** $\tilde{x}_0^{1:V} = p(X, v_{1:V})$
         $\triangleright x_t^{1:V}, \hat{x}_0^{1:V}, \tilde{x}_0^{1:V}, \varepsilon^{1:V} \in \mathbb{R}^{V \times L \times J \times 2}$
7:      **Sample noise:** $\varepsilon^{1:V} \xleftarrow{\text{project}} \varepsilon_{3D} \sim \mathcal{N}(0, I)$
8:      **Denoising step:** $x_{t-1}^{1:V} = \frac{\beta_t \sqrt{\bar{\alpha}_{t-1}}}{1 - \bar{\alpha}_t} x_t^{1:V} + \frac{(1 - \bar{\alpha}_{t-1})\sqrt{\bar{\alpha}_t}}{1 - \bar{\alpha}_t} \tilde{x}_0^{1:V} + \frac{\beta_t(1 - \bar{\alpha}_{t-1})}{1 - \bar{\alpha}_t} \varepsilon^{1:V}$
9: **end for**
10: **Output triangulation:** $\underset{X'}{\arg\min} \, ||p(X', v_{1:V}) - x_0^{1:V}||_2^2$

---

## B   Gradient Update Formula

In order to clarify the difference between SDS and our method, we calculate the gradient update formula w.r.t our optimized loss. Denote by $X^{(i)}$ the optimizing motion at iteration $i$. When differentiating our loss w.r.t $X^{(i)}$ we get:

$$\nabla_{X^{(i)}} || p\left(X^{(i)}\right) - \hat{x_0}||_2^2 = \left( p\left(X^{(i)}\right) - \frac{x_t - \sqrt{1 - \bar{\alpha}_t} \epsilon_\phi(x_t)}{\sqrt{\bar{\alpha}_t}} \right) \frac{\partial p}{\partial X^{(i)}} \tag{3}$$

which is clearly differers from $\nabla \mathcal{L}_{\text{SDS}}$. Let us observe substituting our $x_t$ sampling with a simple forward diffusion: $x_t = \sqrt{\bar{\alpha}_t} p\left(X^{(i-1)}\right) + \left(\sqrt{1 - \bar{\alpha}}\right) \varepsilon$ - as used in DreamFusion. (This formulation is also analyzed in HIFA (Zhu & Zhuang, 2023)):

$$\nabla_{X^{(i)}} || p\left(X^{(i)}\right) - \hat{x_0}|| = \left( p\left(X^{(i)}\right) - \frac{x_t - \sqrt{1 - \bar{\alpha}_t} \epsilon_\phi(x_t)}{\sqrt{\bar{\alpha}_t}} \right) \frac{\partial p}{\partial X_i} = \tag{4}$$

$$\left( p\left(X^{(i)}\right) - \frac{\sqrt{\bar{\alpha}_t} p\left(X^{(i-1)}\right) + \sqrt{1 - \bar{\alpha}_t}\varepsilon - \sqrt{1 - \bar{\alpha}_t}\epsilon_\phi(x_t)}{\sqrt{\bar{\alpha}_t}} \right) \frac{\partial p}{\partial X_i} = \tag{5}$$

$$\left( p\left(X^{(i)}\right) - p\left(X^{(i-1)}\right) + \frac{\sqrt{1 - \bar{\alpha}_t}}{\sqrt{\bar{\alpha}_t}} (\varepsilon - \epsilon_\phi(x_t)) \right) \frac{\partial p}{\partial X_i} \tag{6}$$

If we observe the first iteration of optimization, we have $X^{(i)} = X^{(i-1)}$ so we get:

$$\nabla_X || p(X) - \hat{x_0}||_2^2 = \frac{\sqrt{1 - \bar{\alpha}_t}}{\sqrt{\bar{\alpha}_t}} (\varepsilon - \epsilon_\phi(x_t)) \frac{\partial p}{\partial X} \tag{7}$$

This shows that SDS loss is a special case of our loss when sampling $x_t = \sqrt{\bar{\alpha}_t} p\left(X^{(i-1)}\right) + \left(\sqrt{1 - \bar{\alpha}}\right) \varepsilon$ (where $\varepsilon \sim \mathcal{N}(0, I)$), and applying only a single optimization step (after the first step, $X^{(i)} \neq X^{(i-1)}$).

## C   Theorems

**Theorem 1.** *Let* $\varepsilon = \begin{pmatrix} x_\varepsilon \\ y_\varepsilon \\ z_\varepsilon \end{pmatrix} \sim \mathcal{N}(0, I_{3\times 3})$ *and let* $P \in \mathbb{R}^{2 \times 3}$ *be an orthogonal projection matrix, then* $P \cdot \varepsilon \sim \mathcal{N}(0, I_{2\times 2})$.

*Proof.* First, $P \cdot \varepsilon$ has a normal distribution as a linear combination of normal variables.
In addition, $\mathbb{E}[P \cdot \varepsilon] = P \cdot \mathbb{E}[\varepsilon] = 0$.

Now we will prove that $\text{Var}\left[P\cdot\varepsilon\right]=I_{2\times2}$:

Denote $O=\begin{pmatrix}1 & 0 & 0\\ 0 & 1 & 0\end{pmatrix}$ then we know that $P=O\cdot P'$ where $P'$ is a rotation matrix, i.e. $P'\cdot(P')^T=I_{2\times2}$. Then

$$P\cdot P^T=(O\cdot P')(O\cdot P')^T=O\cdot\overbrace{(P'P'^T)}^{I}O^T=OO^T=I$$

Furthermore, $\mathbb{E}\left[\varepsilon\cdot\varepsilon^T\right]=\mathbb{E}\left[\varepsilon\cdot\varepsilon^T\right]-\overbrace{\mathbb{E}\left[\varepsilon\right]\mathbb{E}\left[\varepsilon\right]^T}^{0}=\text{Var}\left[\varepsilon\right]=I$.

Therefore:

$$\text{Var}\left[P\cdot\varepsilon\right]=\mathbb{E}\left[(P\cdot\varepsilon)(P\cdot\varepsilon)^T\right]-\overbrace{\mathbb{E}\left[P\cdot\varepsilon\right]}^{0}\cdot\overbrace{\mathbb{E}\left[P\cdot\varepsilon\right]^T}^{0}=$$

$$\mathbb{E}\left[P\cdot\varepsilon\cdot\varepsilon^T\cdot P^T\right]=P\cdot\overbrace{\mathbb{E}\left[\varepsilon\cdot\varepsilon^T\right]}^{I}\cdot P^T=P\cdot P^T=I$$

$\square$

**Theorem 2.** *Let $X\in\mathbb{R}^3$, and denote by $p_{orth}(X),p_{pers}(X)$ the orthographic and perspective projections of $X$ to the same view, respectively. Then $p_{orth}(X)-p_{pers}(X)$.*

*Proof.* First, denote the rotation matrix that corresponds to the view by $R\in\mathbb{R}^{3\times3}$ and the distance by $d$. Now, also denote $R_{xy}=\begin{pmatrix}1 & 0 & 0\\ 0 & 1 & 0\end{pmatrix}\cdot R$, $R_z=(0\quad 0\quad 1)\cdot R$. Then

$$p_{\text{orth}}(X)=R_{xy}\cdot X, p_{\text{pers}}(X)=\frac{R_{xy}\cdot X}{d+R_z\cdot X}\cdot d$$

So

$$p_{\text{orth}}(X)-p_{\text{pers}}(X)=R_{xy}\cdot X-\frac{R_{xy}\cdot X}{d+R_z\cdot X}\cdot d=\frac{R_{xy}\cdot X\cdot(d+R_z\cdot X-d)}{d+R_z\cdot X}=$$
$$\frac{R_{xy}\cdot X\cdot R_z\cdot X}{d+R_z\cdot X}$$

Assume $\|X\|_2\le1$, then $\frac{R_{xy}\cdot X\cdot R_z\cdot X}{d+R_z\cdot X}\le\frac{1}{d-1}$. $\square$

## D  DYNAMIC VIEW-POINT SAMPLING

Keeping the optimized views constant could theoretically lead to overfitting a motion to the optimized views, while novel views might have a lower quality. Note that this problem arises only at a lower number of views (¡5). For this reason, we suggest a way to re-sample the viewing-points: After every step, we can save $X^{(i)}$ and the 3D noise sample used $\epsilon_{3D}^{(i)}$. When trying to sample $x_t$ for a newly sampled view $v$ we can then take all $X^{(0)},...,X^{(T-t)}$, and all $\epsilon_{3D}^{(0)},...,\epsilon_{3D}^{(T-t)}$ and project them to view $v$. We can then apply a sampling loop using the projections, just like we did in the original algorithm. We observe that in our setting, this method does not lead to significant improvement so we present it as an optional addition.

