# OpenReview forum: "MAS: Multi-view Ancestral Sampling for 3D motion generation using 2D diffusion"
_ICLR.cc/2024/Conference — ICLR 2024 Conference Withdrawn Submission_

### Official Review · Reviewer_XKj7 · 2023-10-31

**Soundness:** 1 poor
**Presentation:** 2 fair
**Contribution:** 2 fair
**Rating:** 3
**Confidence:** 4

**Summary:**

This paper is about 3D lifting of given 2D features points by a diffusion model. The key points are connected by a kinematic chain. According to the explanation im the paper, the authors explain that 3D models corresponding to multiple time instances are triangulated. The triangulated poses are backprojected into 2D. The diffusion noise is generated in 3D but projected by orthographic projection into the images. It is added to the backprojected 2D joint positions. The authors present results on 2 publicly available datasets and one created by them.

**Strengths:**

The idea to use a diffusion model to generate 3D joint positions from 2D points is interesting.

**Weaknesses:**

I am wondering how the triangulation is done. I find the explanation given in the paper deficient with respect to this problem. Given the sequence of 2D joint positions per image, the authors only explain that they triangulate the 2D points to obtain 3D. For rigid motion, this idea is reasonable. However, the motion is non-rigid. Assuming that the camera does not move between 2 images, corresponding 2D projections can stem from different 3D points, in other words. If I understand the explanation given on page 6 (motion block consistency and triangulation) correctly, then only a single denoised version is created for each input image. Am I missing something?

The idea to use virtual 3D views is not new. In the GAN/adversarial era, this was a common technique, see, for instance,
Unsupervised Adversarial Learning of 3D Human Pose from 2D Joint Locations, Kudo et al, 2018

The experimental evaluation is deficient. There are many other works doing 3D reconstruction of animals. 2 papers from this year:
* Learning Articulated Shape with Keypoint Pseudo-Labels from Web Images, Stathopoulos er al., CVPR’23
* Farm3D: Learning Articulated 3D Animals by Distilling 2D Diffusion, Jakob eat al., arXiv’23
There are many more papers from previous years. Neither are any of those works cited nor are they used for comparison.

**Questions:**

* What does the little lock symbol in Fig 2 mean?

---

### Official Review · Reviewer_ZnEM · 2023-11-01

**Soundness:** 2 fair
**Presentation:** 3 good
**Contribution:** 3 good
**Rating:** 5
**Confidence:** 4

**Summary:**

The paper proposes an unconditional 3D motion generation method by only leveraging 2D diffusion priors. The key technique is
multi-view ancestral sampling (MAS). Built upon original diffusion de-noising, the paper introduces multi-view de-noising in a single time-step. To keep consistency across views, the paper performs 3D triangulation by fusing different view predictions and project back to each view afterwards. The paper also presents to sample 3D Gaussian noise and project to each view to maintain 3D consistency. MAS achieves good generation performance in terms of consistency, diversity and quality compare against existing baselines.

**Strengths:**

- **Good motivation**. When target 3D data is hard to acquire, the paper introduces a good solution to learn 3D motions from 2D priors.

- **Simple yet efficient method**. The method adopts explicit triangulation and 3D noise sampling to guarantee multi-view consistency. The results demonstrate the effectiveness when evaluated on side-views. The generation is also fast and light-weight.

- **Qualitative improvement**. The papers shows qualitative motion improvement against previous-work on NBA player videos.

**Weaknesses:**

- **Evaluation on naturalness of motion**. First, there is no evaluation on the naturalness of the generated motion. From the provided video, it seems the motion of the horse sometime is unnatural. In the human-ball case, there is always no contact between the ball and the human. It is recommend to perform some user-study to compare the generated motion's naturalness and quality against GT and other baselines.


- **Evaluation on dataset where 3D GT is available**. I understand the goal is to perform generation on less common scenes where 3D GT is unavailable, but perform direct evaluation on 3D is still useful and necessary to benchmark the performance and evaluate 3D consistency. It is recommend to see the performance gap to GT in 3D on any of existing 3D human or animal datasets.


- **Failures in 3D triangulation**. One assumption the paper made about 2D diffusion is that the trained model could generalize to multi-views. But in reality, some views are less common compare against other views, the obtained prediction from 2D estimator could also be incorrect. These could potentially caused huge error in the triangulation step. I am curious what the performance will be like under some extreme cases, and also is there any common failure modes of the method.


- **Typos and unreferenced figures**.  "Although these methods are not generative per s" the sentence seems incomplete, and figure 6 in the same paragraph should be figure 4.

**Questions:**

- **How to obtain 3D shapes in visualization**. The method output 3D skeleton but the visualization shows the 3D shapes. The author should tell  what models they are using to get those animatable models from 3D skeleton, especially for non-humanoid categories (i.e. horse).


- **What is the performance if we use perspective projection**.  I read the theorem in the paper but I am still eager to see some quantitative comparison between using perspective projection and orthogonal projection in 3D noise sampling.


- **Comparison with pose lifting**. For method like ElePose, is the number reported for the fine-tuned model on the collected in-the-wild data or the pre-trained model. Given it is unsupervised, the model should be fine-tuned on those videos and evaluate the performance.

---

### Official Review · Reviewer_8H4R · 2023-11-02

**Soundness:** 2 fair
**Presentation:** 2 fair
**Contribution:** 2 fair
**Rating:** 5
**Confidence:** 3

**Summary:**

The paper introduces a generative approach to generate realistic 3D motions of complex articulated object classes (e.g. humans, horses). using diffusion models. The key novelty of the work is that the method is trained on 2D video sequences only, unlike previous work that relied on 3D motion sequences. This is very powerful, because video data is readily available, while 3D motion capture sequences are expensive to generate.

As pre-processing, the method extracts object tracks and SOTA 2D pose estimation to recover the 2D position of the joints of the articulated structures in the video (2D motions). It then trains a generative model of 2D motions on the 2D motions from the training dataset using a recent diffusion model for motion (MDM). Lastly, it learns a de-noising generative model for 3D motions on the latent also using a diffusion model. Intuitively, this is trained as follows: given a 2D motion in the training set, the 2D motion diffusion model (MDM) is used to generate multiple 2D views of this single 2D motion, and a 3D motion is fit to these multiple views via triangulation; this is then re-projected in 2D producing a version of the 2D views that is more geometrically consistent, on which in turn the 2D diffusion model is re-applied (and it keeps going for a fixed number of iterations).

The generative power of the full method is evaluated on three different object classes (horse jumping, basketball players, gymnasts)  using metrics traditionally used for evaluating unconditional generative models (Frechet inception distance, diversity precision, recall). Results includes comparison against related work, that, unlike this paper, was designed to be trained on 3D data.

**Strengths:**

1. The key novelty of the work is that the method is trained on 2D video sequences only, unlike previous work that relied on 3D motion sequences. This is very powerful, because video data is readily available, while 3D motion capture sequences are expensive to generate.
2. The main technical contribution of the paper, i.e. the ancestral sampling with 3D consistency, seems to be novel and have merit
3. The method produces impressive qualitative results
4. The authors will release a new dataset of 60k+ videos

**Weaknesses:**

1. I found the quantitative evaluation provided insufficient for several reasons. The method is only evaluated on its unconditional generative power in 2D space, and I am not convinced that this is sufficient to assess the realism and diversity of the generated 3D motions. In principle, it is possible to generate 3D motions that are quite unrealistic but produce reasonable 2D motions. While the the qualitative results provide good evidence about the quality of generated 3D motions, I think evaluation in 3D is needed. For example, why can the method not be assessed on datasets like Human3.6M, e.g. by comparing the distribution of the generated 3D motions to the  distribution. of ground-truth 3D motions in the dataset? Moreover, while it is true that the metrics used are standard, the methods compared against typically report other metrics like 3D evaluation or human judgment for conditional generation of 3D motion.  Lastly, I am not convinced the comparison to MotionBert and ElePose is fair, as I am assuming that either method was not fine-tuned on the target dataset of gymnasts, NBA players, etc.
2.I could not understand important aspects of the evaluation procedure that made it hard for me to interpret the results, ranging from what dataset is used for evaluation, to how the related work is trained on this data (see details in the questions session).
3. I could not understand some important technical details of MAS - this might be in part due to me not being proficient in diffusion models. In particular, I could not understand how the 2D diffusion model (MDM) is used to generate multiple views given an input 2D motion, i.e. is there any step enforcing that the samples from MDM come from different views, or are these just random samples and the iterative process causes them to converge on plausible 2D rendering from each view? This seems an importatn aspect, and I am struggling to understand how this could converge to something reasonable


Minor: There is no Figure 6, although it is mentioned in the last sentence in Section 6 (I assume it's a type and it should be figure 4)

**Questions:**

1.  Why can the method not be assessed on datasets like Human3.6M, e.g. by comparing the distribution of the generated 3D motions to the  distribution. of ground-truth 3D motions in the dataset?
2. What dataset is used for quantitative evaluation? Is it the basketball player, gymnast, horse jumping dataset? If yes, why are results not broken down by object class?
3. In. Table 2, MAS achieves significant less recall than MDM (0.6 vs 0.8). Can you comment on why it is the case and what could be causing it?
4. Are related work methods trained/fine-tuned on the evaluation dataset? I am assuming no, since these baselines explicitly need 3D data for training, so neither MotionBert nor ElePose can be trained on the NBA players/gymnasts/horse dataset. Hence, how can the comparison be fair given that MAS is trained on this dataset? Can you comment on this point?
5. Why aren't there qualitative results for basketball players? In general, are the qualitative results provided samples from the model, or "de-noised" results for 2D samples in the dataset? If yes, can they be shown side by side?
6. Can you help me understand how MDM is used to generate multiple views for each 2D motion, in a way that the process can actually converge to something reasonable (see my point 3 in Weaknesses)
7. Can you help me understand better the relationship between MAS and a standard diffusion model? Can MDM be seen as the forward process, why the 3D consistency check as the reverse process? Or is this incorrect?

---

### Author Response · Authors · 2023-11-22
**General response**

Dear reviewers, we would like to thank you for taking the time to read and evaluate our paper. Your remarks and insights will help us refine our paper and make it clearer to future readers. We feel that our method and its merits were not understood properly:
* **MAS is a generative approach** - not a pose-lifting method. It uses a 2D diffusion model to transform random noise into a 3D motion.
* The 2D diffusion model does not natively generate multiple views of a given 2D motion - as reviewer 8H4R implies. Instead, our 3D noise and consistency block ensure that the generated 2D motions describe the same 3D motion.
* **MAS is an inference algorithm** on top of the 2D diffusion models, and does not train a 3D diffusion model.
* The triangulation stage is performed by taking multiple 2D motion predictions generated by the model, and directly optimizing the joint locations of a 3D motion to fit all 2D motions upon projection. MAS does not attempt to triangulate the motion by using different frames - as reviewer XKj7 suggests.
In our next revision, we will make the method presentation clearer following your impressions.